# RETHINKING HALLUCINATIONS: CORRECTNESS, CONSISTENCY, AND PROMPT MULTIPLICITY

**Prakhar Ganesh**
McGill University & Mila
`prakhar.ganesh@mila.quebec`

**Reza Shokri**
National University of Singapore
`reza@comp.nus.edu.sg`

**Golnoosh Farnadi**
McGill University & Mila
`farnadig@mila.quebec`

## ABSTRACT

Large language models (LLMs) are known to "hallucinate" by generating false or misleading outputs. Hallucinations pose various harms, from erosion of trust to widespread misinformation. Existing hallucination evaluation, however, focuses only on "correctness" and often overlooks "consistency", necessary to distinguish and address these harms. To bridge this gap, we introduce *prompt multiplicity*, a framework for quantifying consistency through prompt sensitivity. Our analysis reveals significant multiplicity (over $50\%$ inconsistency in benchmarks like Med-HALT), suggesting that hallucination-related harms have been severely underestimated. Furthermore, we study the role of consistency in hallucination detection and mitigation. We find that: (a) detection techniques capture consistency, not correctness, and (b) mitigation techniques like RAG can introduce additional inconsistencies. By integrating prompt multiplicity into hallucination evaluation, we provide an improved framework of potential harms and uncover critical limitations in current detection and mitigation strategies.

## 1 INTRODUCTION

Large language models (LLMs) have been widely adopted, excelling in numerous tasks across diverse domains (Guo et al., 2023; Kasneci et al., 2023; Etsenake & Nagappan, 2024). Despite their growing use, LLMs suffer from a critical limitation: generation of false, nonsensical or misleading outputs, studied under the umbrella of hallucinations. The term "hallucinations" has evolved over the years, shifting from positive use in computer vision (Baker & Kanade, 2000; Hsu et al., 2010) to a predominantly negative association in natural language processing (NLP) (Karpathy, 2015; Huang et al., 2023; Ji et al., 2023; Zhang et al., 2023). It is commonly defined as *'generated content that is nonsensical or unfaithful to the provided source content'* (Filippova, 2020; Maynez et al., 2020; Zhou et al., 2021).

With growing interest in this field, several benchmarks have been developed to assess hallucination risks in LLMs (Lin et al., 2022; Pal et al., 2023; Muhlgay et al., 2024; Lattimer et al., 2023; Dong et al., 2024; Li et al., 2023; Hong et al., 2024). Unfortunately, a critical aspect of evaluation still remains largely overlooked—consistency across prompt variations. While prompt sensitivity has been extensively studied in LLM benchmarking (Sclar et al., 2023; Shi et al., 2023; Pezeshkpour & Hruschka, 2024; Alzahrani et al., 2024; Mizrahi et al., 2024), it has not received the same attention in hallucination. We argue that this oversight exists because prompt sensitivity literature focuses solely on accuracy variance, and previous works have found that overall accuracies on hallucination

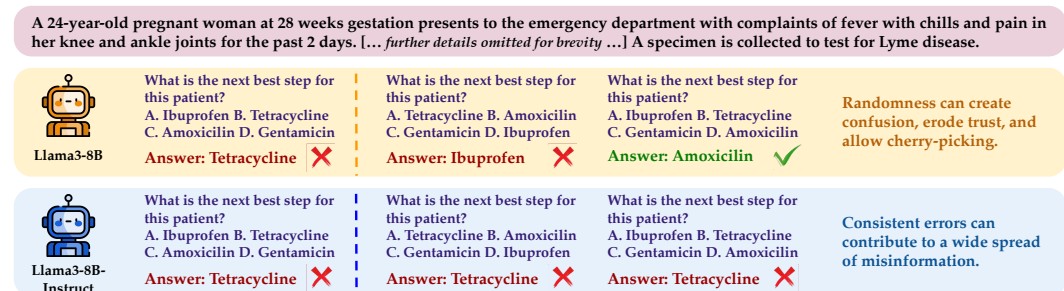

Figure 1: Different harms that are treated the same in existing evaluation. Prompt sensitivity under MCQ options shuffling. Example from the Med-HALT dataset (Pal et al., 2023).

benchmarks remain stable even under paraphrasing (Lin et al., 2022; Hong et al., 2024; Pal et al., 2023).

However, overall accuracy stability can hide the lack of consistency in individual generations. In our work, we formalize consistency in hallucination evaluation through the lens of multiplicity (Marx et al., 2020; Black et al., 2022a), and show that these benchmarks exhibit *high multiplicity*, i.e., the model's response to individual questions changes frequently based on the prompt structure. For instance, while the Med-HALT dataset (Pal et al., 2023) has an accuracy variance of less than $0.5\%$ under changing prompt structure, it showed more than $50\%$ multiplicity (Table 1), i.e., for more than $50\%$ questions the model generates different facts based on the prompt structure, despite stable average accuracy of correct generations.

Leveraging this axis of consistency quantified using multiplicity, we provide a more nuanced decomposition of hallucination errors. We find that existing evaluations solely based on correctness can hide differences and underestimate the real risks of hallucinations. Consider two well-known datasets, TruthfulQA (Lin et al., 2022) and Med-HALT (Pal et al., 2023), with similar accuracies ($25-30\%$). However, we show that models make very distinct errors on these datasets, with TruthfulQA dominated by consistent yet factually incorrect generations, and Med-HALT dominated by randomness and inconsistency (Figure 3). Moreover, we find consistently correct generations on these datasets are far lower than their accuracies ($15-20\%$), highlighting the overestimation of model capabilities.

The distinction between various errors helps us classify the real harms and plays a pivotal role in shaping discussions on addressing hallucinations. We position existing detection and mitigation techniques within our framework, identifying limitations and improving our understanding of their effectiveness. We connect detection techniques with multiplicity, demonstrating that they detect consistency, not correctness (Figure 4). Thus, there is a misalignment between detection, which aims to detect consistency, and the benchmarks, which are instead designed to evaluate correctness. Finally, we move to mitigation, showing that while the introduction of components like retrieval-augmented generation (RAG) (Ram et al., 2023) reduce overall hallucination rates, these improvements can hide a new inconsistency due to prompt sensitivity of the retrieval itself (Figure 5).

Our key contributions are:

- **Prompt multiplicity in LLM hallucination evaluation:** We formalize consistency in hallucination evaluation by defining *prompt multiplicity*, leveraging existing tools from the multiplicity literature (§3.2). We highlight severe prompt multiplicity across six different benchmarks and 16 different models (from six model families), undermining the reliability of existing evaluation frameworks in quantifying the true harms of hallucinations (§4.2).
- **An improved taxonomy for benchmarking:** We propose a refined taxonomy for hallucination benchmarking by quantifying *'prompt-agnostic vs prompt-sensitive'* (Yin et al., 2024) and *'randomness'* (Venkit et al., 2024), through the lens of prompt multiplicity (§3.3). We also highlight the advantages of our framework in assessing real-world risks and illustrate several dataset-specific trends to map future progress in various domains (§4.3).
- **Hallucination detection and mitigation under prompt multiplicity:** We establish that existing detection techniques do not detect correctness, but instead detect a different axis of hallucination

evaluation, i.e., consistency (§5.1), highlighting the disconnect between methods and benchmarks. Finally, we show that mitigation techniques like RAG are also affected by prompt sensitivity, and thus introduce additional inconsistencies (§5.2).

## 2 RELATED WORK

In our work, we propose a new framework to improve existing LLM hallucination benchmarks by examining how prompt sensitivity influences hallucination evaluation, studied through the lens of multiplicity. This section explores related work across these key areas.

**LLM Hallucination Benchmarks.** Hallucinations in LLMs have garnered significant interest, with extensive work on categorization, evaluation, detection, and mitigation (Huang et al., 2023; Ji et al., 2023; Wang et al., 2023; Zhang et al., 2023; Tonmoy et al., 2024). Various hallucination benchmarks have been developed, with a variety of task settings like multiple-choice questions (MCQs) (Petroni et al., 2019; Lin et al., 2022; Pal et al., 2023; Muhlgay et al., 2024), summarization (Lattimer et al., 2023; Dong et al., 2024), generation (Li et al., 2023), etc. More recently, Hong et al. (2024) combined multiple benchmarks into a single leaderboard for a holistic evaluation of hallucinations. We propose a new evaluation framework that incorporates consistency, and can be extended to any existing benchmark.

**Prompt Sensitivity in LLMs.** Prompt sensitivity studies in LLMs have revealed that even minor changes to the input or the prompt structure can impact model behaviour (Lu et al., 2022; Shi et al., 2023; Sclar et al., 2023; Voronov et al., 2024). Recent research has also heavily focused on the MCQ format, widely used in LLM evaluations, finding that changes to the order or representation of choices can also affect model accuracy (Zheng et al., 2023; Pezeshkpour & Hruschka, 2024; Alzahrani et al., 2024; Polo et al., 2024; Mizrahi et al., 2024). However, literature on prompt sensitivity in hallucinations remains limited. While Lin et al. (2022); Pal et al. (2023); Hong et al. (2024) have performed small-scale ablation studies in their work to study the impact of prompt paraphrasing on hallucination benchmarks, they found stable overall accuracy trends and thus did not explore question-level behaviour of hallucinations. We aim to address this critical gap in the literature.

**Multiplicity.** Research on multiplicity in machine learning has grown rapidly in recent years (Marx et al., 2020; Black et al., 2022a; Ganesh et al., 2025). A key subtopic, predictive multiplicity (Marx et al., 2020), refers to the existence of multiple models with similar overall accuracy but different individual-level predictions. We extend the notion of multiplicity to what we call *prompt multiplicity* in LLMs. Specifically, we study how competing prompt structures can yield similar benchmark accuracy while generating different individual-level answers. We use the multiplicity framing to take advantage of the existing literature.

## 3 HALLUCINATIONS: INCORRECT KNOWLEDGE OR RANDOMNESS?

In the existing literature, any plausible-sounding but factually incorrect or nonsensical text generated by a model is termed a "hallucination" (Venkit et al., 2024; Ji et al., 2023). This covers a wide range of model behaviour, from "incorrect knowledge" to "randomness" (Venkit et al., 2024). However, hallucinations as factually incorrect knowledge embedded in the model[1] due to outdated information, flawed data sources, biases, or myths present in the training data scraped from the web (Huang et al., 2023; Lin et al., 2022) form a distinct category from hallucinations as random but plausible-sounding generations, sometimes referred to as 'confabulations' (Millidge, 2023; Farquhar et al., 2024).

In this section, we begin by discussing the two broad categories of harm from hallucinations, emphasizing the key distinction between them, i.e., "consistency". As existing benchmarks do not measure consistency, to address this gap, we draw from the multiplicity literature and formalize

---

[1]We use *knowledge embedded in the model* to refer to prompt-agnostic knowledge (Yin et al., 2024). Despite the tension between 'knowledge in LLMs can be difficult to extract' (Gekhman et al., 2025; Yin et al., 2024) and 'LLMs can be forced to generate any correct or incorrect fact' (Yao et al., 2023), for the sake of understanding the potential harms of LLM hallucinations, we argue that any information consistently repeated by the model can be considered knowledge embedded in the model.

*prompt multiplicity*. Finally, we use this new axis of consistency to provide a more refined taxonomy for hallucination evaluation, directly mapping to several aspects of hallucinations that have been previously discussed but never explicitly quantified.

### 3.1 HARMS FROM LLM HALLUCINATIONS AND THE ROLE OF CONSISTENCY

The harm caused by hallucinations in language models depends on several factors, including the model's use case, the user's level of trust in the model, and their expertise, among others (Venkit et al., 2024; Elsayed, 2024; Bender et al., 2021). One way to broadly categorize these harms is by evaluating the consistency of hallucinations. To understand this distinction, consider the example in Figure 1. Two models, Llama3-8B and Llama3-8B-Instruct (Dubey et al., 2024), make the same error on the Med-HALT dataset (Pal et al., 2023). Without accounting for consistency, both errors appear identical and are labelled as "hallucinations" in existing benchmarks. However, by testing the models multiple times with various equivalent prompts (here, shuffling the order of MCQ options), we uncover a key difference.

Llama3-8B exhibits inconsistency, i.e., it selects different answers depending on the prompt variation. This unpredictability can erode user trust, confuse even an expert working alongside the model, and introduce the risk of cherry-picking certain responses. In contrast, Llama3-8B-Instruct consistently provides the same incorrect answer. It repeatedly identifies Tetracycline as its choice, which is unfortunately the wrong antibiotic in this situation, as unlike Amoxicilin, it has known risks for pregnant women. This consistency in hallucination creates a different harm: rather than hiding uncertainty with confident generations, the model is propagating misinformation. The two categories of harm can be defined as follows.

**Harms due to randomness.** Hallucinations can arise when the model is uncertain about the correct answer or is confidently guessing. Such hallucinations would be likely *prompt-sensitive* (Yin et al., 2024), i.e., the response can vary based on the prompt. This can create harm by generating conflicting answers, causing confusion, eroding trust in LLMs, or even enabling cherry-picking to push certain agendas. Detecting these errors requires quantifying the uncertainty of LLM generations (Vashurin et al., 2024; Savage et al., 2024).

**Harms due to incorrect knowledge embedded in the model.** Hallucinations can also occur when LLMs encode incorrect or partial knowledge, misconceptions, or myths, from the training data. These can mislead users in critical contexts or contribute to a wider spread of misinformation (Venkit et al., 2024). Such hallucinations are likely *prompt-agnostic* (Yin et al., 2024), i.e., the model consistently generates the same incorrect response. These errors cannot be addressed by simply measuring uncertainty, and might require filtering unreliable training data or fact-checking the generated sentences using external knowledge.

Consistency thus plays an important role in understanding the causes, impact, and effective strategies to address hallucinations. An inconsistent hallucination stems from randomness, while a consistent one may reveal flawed data sources. Thus, by incorporating consistency into hallucination evaluation, we can develop a more nuanced understanding of these risks.

### 3.2 DEFINING PROMPT MULTIPLICITY

Literature on prompt sensitivity focuses primarily on accuracy stability (Sclar et al., 2023; Voronov et al., 2024; Mizrahi et al., 2024), which unfortunately does not capture the question-level consistency concerns discussed above. Interestingly, this exact problem lies at the heart of the field of predictive multiplicity (Marx et al., 2020; Black et al., 2022a). Predictive multiplicity refers to the existence of multiple models achieving similar accuracy, yet exhibiting distinct behaviour in individual predictions. Drawing from this, we propose *prompt multiplicity*, the idea that different prompt structures can achieve similar average hallucination accuracy, yet produce distinct behaviour for individual questions. This framework allows us to capture "consistency" in LLM hallucination evaluation.

In our paper, we use MCQ-style benchmarks, where the goal is to select the factually correct continuation from a set of options (more details in §4.1). Each question in the benchmark $\mathbf{x}_k \in \mathbf{X}$ is first formatted using a prompt structure $\mathbf{p}^i$, which may involve modifications like prefixing demonstrations to the question, adding instructions, etc., before it is fed to a model $\mathbf{G}$. We use the notation

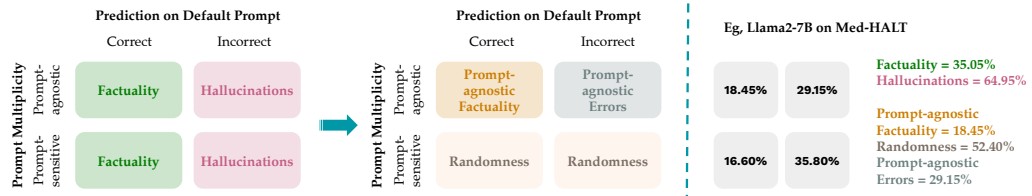

Figure 2: The mapping from existing terms like *"hallucinations"* and *"factuality"* to a more nuanced taxonomy of *"prompt-agnostic factuality"*, *"prompt-agnostic errors"*, and *"randomness"*.

$\mathbf{G}(\mathbf{p}^i(\mathbf{x}_k))$ to indicate the final MCQ choice of the model $\mathbf{G}$ for a question $\mathbf{x}_k$ with prompt structure $\mathbf{p}^i$. Building on Marx et al. (2020), we define prompt multiplicity and ambiguity under a set of prompt structures $\mathbb{P} = [\mathbf{p}^1, \mathbf{p}^2, ..., \mathbf{p}^r]$ as follows:

**Definition 1** (Prompt Multiplicity). *Given a model $\mathbf{G}$, a benchmark $\mathbf{X}$, and a set of prompt structures, $\mathbb{P} = [\mathbf{p}^1, \mathbf{p}^2, ..., \mathbf{p}^r]$, the benchmark is said to exhibit prompt multiplicity if $\exists \mathbf{p}^i, \mathbf{p}^j \in \mathbb{P}$ such that $\mathbf{G}(\mathbf{p}^i(\mathbf{x}_k)) \neq \mathbf{G}(\mathbf{p}^j(\mathbf{x}_k))$ for some question $\mathbf{x}_k \in \mathbf{X}$.*

**Definition 2** (Ambiguity). *Given a model $\mathbf{G}$, a benchmark $\mathbf{X}$, and a set of prompt structures, $\mathbb{P} = [\mathbf{p}^1, \mathbf{p}^2, ..., \mathbf{p}^r]$, ambiguity is the proportion of questions in the benchmark that can output different choices depending on the prompt structure,*

$$Ambiguity = \frac{1}{n} \sum_{k=1}^{n} \max_{\mathbf{p}^i, \mathbf{p}^j \in \mathbb{P}} \mathbb{1}[\mathbf{G}(\mathbf{p}^i(\mathbf{x}_k)) \neq \mathbf{G}(\mathbf{p}^j(\mathbf{x}_k))] \tag{1}$$

While we can use ambiguity to quantify the severity of multiplicity in a benchmark, we also need a metric to define "consistency" for each question separately. For this, we turn to Cooper et al. (2024), and define self-consistency as,

**Definition 3** (Self-consistency). *Given a model $\mathbf{G}$, a question $\mathbf{x}_k \in \mathbf{X}$, and a set of prompt structures, $\mathbb{P} = [\mathbf{p}^1, \mathbf{p}^2, ..., \mathbf{p}^r]$, self-consistency is the probability of getting the same output choice from two randomly chosen prompt structures $\mathbf{p}^i, \mathbf{p}^j \sim \mathbb{P}$,*

$$SC_{x_k} = 1 - Prob_{\mathbf{p}^i, \mathbf{p}^j \sim \mathbb{P}}[\mathbf{G}(\mathbf{p}^i(\mathbf{x}_k)) \neq \mathbf{G}(\mathbf{p}^j(\mathbf{x}_k))] \tag{2}$$

### 3.3 Mapping Prompt Multiplicity to Hallucination Evaluation

Building on the definitions above, we now introduce a new axis of evaluation in our framework, "consistency". While existing benchmarks only divide the evaluations along the axis of correctness, we argue that incorporating consistency can provide more nuance to the discussion and quantify various forms of harm. We use the self-consistency metric (Definition 3) to categorize questions along the consistency axis into prompt-sensitive and prompt-agnostic, adopted from Yin et al. (2024) and defined as follows:

**Definition 4** (Prompt-sensitive). *A question $\mathbf{x}_k \in \mathbf{X}$ is considered prompt-sensitive if its self-consistency score $SC_{x_k}$ is below a given threshold $\tau$,*

$$Prompt\text{-}sensitive \Leftarrow \mathbb{1}[SC_{x_k} < \tau] \tag{3}$$

**Definition 5** (Prompt-agnostic). *A question $\mathbf{x}_k \in \mathbf{X}$ is considered prompt-agnostic if its self-consistency score $SC_{x_k}$ is equal to or above a given threshold $\tau$,*

$$Prompt\text{-}agnostic \Leftarrow \mathbb{1}[SC_{x_k} \geq \tau] \tag{4}$$

We use $\tau = 0.8$ throughout the paper, unless otherwise specified.

**A refined evaluation terminology:** We argue that factually correct generations that are prompt-sensitive, despite being accurate for the default benchmark prompt structure, should be treated with the same level of caution as factually incorrect prompt-sensitive generations. In other words, if the generation of factually incorrect information is highly dependent on the prompt structure, it should be categorized as **randomness**, irrespective of whether this randomness happens to produce the

correct output for the default prompt structure of the benchmark, as it possesses the same risk of generating a factually incorrect sentence for a different prompt structure. Moreover, we propose to use the term **prompt-agnostic factuality** and **prompt-agnostic errors** to describe *prompt-agnostic* generations.

Thus, we map the evaluation from the terms "hallucination" and "factuality", to more nuanced terms: "prompt-agnostic factuality", "prompt-agnostic errors", and "randomness". Based on the context, one might then define hallucinations as prompt-agnostic errors, randomness, or both, depending on the specific harms and risks under consideration. A visual representation of this framework and the mapping is shown in Figure 2.

## 4 PROMPT MULTIPLICITY IN LLM HALLUCINATION BENCHMARKS

We now turn to the empirical results. We first highlight severe multiplicity in hallucination benchmarks, despite stable accuracy. Next, we map the evaluation to our framework, revealing how existing benchmarks underestimate potential real-world harm caused by hallucinations. We conclude with dataset-specific trends and takeaways.

### 4.1 EXPERIMENT SETUP

**Datasets.** We use the following factuality hallucination benchmarks: Wiki-FACTOR (Muhlgay et al., 2024), Med-HALT (Pal et al., 2023), TruthfulQA (Lin et al., 2022), TrueFalse (Azaria & Mitchell, 2023), CommonsenseQA (Talmor et al., 2019), and FEVER (Thorne et al., 2018). Details on each dataset are provided in the appendix (§A.1). We primarily focus on TruthfulQA, Wiki-FACTOR, and Med-HALT in the main paper, while other results are delegated to the appendix (§C). We use the perplexity-based evaluation by Muhlgay et al. (2024), where the LLM chooses the best MCQ option based on the length-normalized perplexity.

We decided to stick with only MCQ-style benchmarks for our study, since freeform generation requires additional automated methods to evaluate generated outputs, such as an LLM judge (Lin et al., 2022; Li et al., 2023; Dong et al., 2024)–which can introduce its own errors, biases, and multiplicity (Li et al., 2024; Ye et al.; Panickssery et al., 2024).

**Models.** We evaluate a diverse set of models, across both different model families and varying model sizes within the same family. Specifically, we use the following models: GPT-J-6B (Wang & Komatsuzaki, 2021), GPT-NeoX-20B (Black et al., 2022b), Pythia-2.8B/6.9B/12B (Biderman et al., 2023), Bloom-3B/7.1B (Workshop et al., 2022), Llama2-7B/7B-Chat/13B/13B-Chat (Touvron et al., 2023), Llama3-8B/8B-Instruct (Dubey et al., 2024), and OPT-6.7B/13B/30B (Zhang et al., 2022).

**Prompt Variations.** We simulate prompt variations in a structured manner, wherever possible, as they can be applied uniformly across the dataset. This includes shuffling the order of demonstrations (Lu et al., 2022) (TruthfulQA, FEVER, TrueFalse) or shuffling the order of MCQ options (Zheng et al., 2023; Pezeshkpour & Hruschka, 2024) (Med-HALT, CommonsenseQA). However, the Wiki-FACTOR benchmark does not provide any opportunity for structured variations. Instead, we turn to automated paraphrasing and use a fine-tuned T5 model (Raffel et al., 2020) trained on a paraphrase dataset from ChatGPT (Vorobev & Kuznetsov, 2023a;b). More details on prompt variations are in the appendix (§A.1).

### 4.2 HALLUCINATION BENCHMARKS SHOW HIGH MULTIPLICITY AND UNDERESTIMATE RISKS

Despite low accuracy variance, LLM hallucination benchmarks exhibit severe prompt multiplicity. To illustrate this, we collect the average accuracy, standard deviation, and ambiguity, across different variations, in Table 1 (only the biggest models from each family are shown, the rest are in Table 2). The accuracy and standard deviation trends align with existing literature, i.e., low variance in accuracy. This explains why previous research has largely overlooked prompt sensitivity. However, the ambiguity scores tell a more compelling story, revealing significant prompt multiplicity within these benchmarks. For instance, LLama2-13B-Chat on Med-HALT achieves $\sim 35\%$ accuracy with a standard deviation of only $0.23\%$, potentially signalling stability. Yet, its ambiguity score is $\sim 60\%$,

| | TruthfulQA | | Wiki-FACTOR | | Med-HALT | |
|---|---|---|---|---|---|---|
| | Accuracy (%) | Ambiguity (%) | Accuracy (%) | Ambiguity (%) | Accuracy (%) | Ambiguity (%) |
| GPTNeoX-20B | $20.09_{\pm 1.26}$ | 22.89 | $45.74_{\pm 1.38}$ | 41.95 | $28.98_{\pm 0.43}$ | 52.26 |
| Pythia-12B | $20.31_{\pm 0.89}$ | 19.58 | $42.90_{\pm 0.96}$ | 38.61 | $28.18_{\pm 0.39}$ | 50.07 |
| Bloom-7.1B | $23.18_{\pm 1.32}$ | 18.36 | $35.14_{\pm 0.78}$ | 37.27 | $28.51_{\pm 0.62}$ | 56.40 |
| Llama2-13B-C | $32.97_{\pm 1.03}$ | 21.79 | $50.32_{\pm 1.06}$ | 47.09 | $34.84_{\pm 0.42}$ | 60.54 |
| Llama3-8B-I | $39.34_{\pm 0.75}$ | 17.14 | $48.39_{\pm 1.19}$ | 42.32 | $34.55_{\pm 0.23}$ | 31.03 |
| OPT-30B | $22.55_{\pm 0.90}$ | 23.38 | $43.58_{\pm 0.91}$ | 41.35 | $28.32_{\pm 0.42}$ | 50.00 |

Table 1: High ambiguity across a wide range of model families and benchmarks.

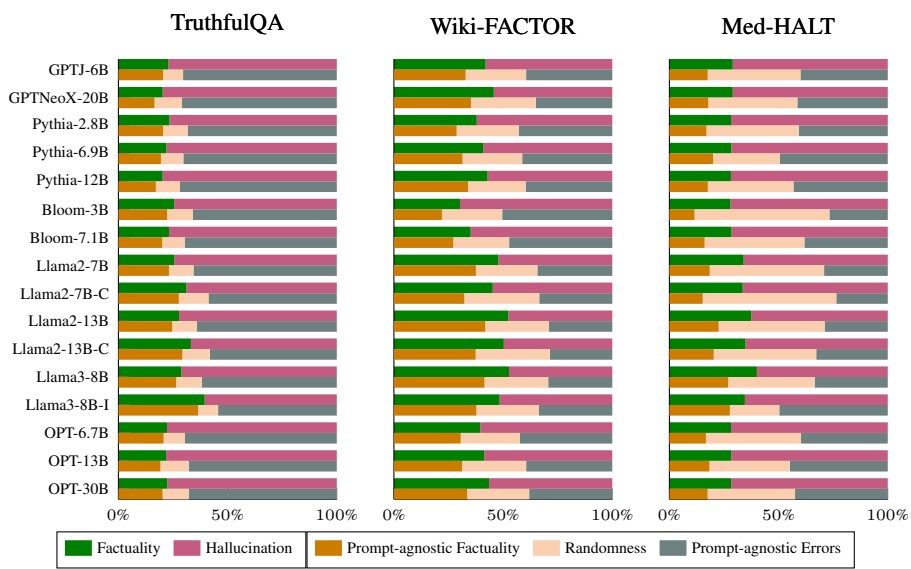

Figure 3: Existing LLM hallucination evaluation terminology vs our framework.

i.e., the model changes the generated fact for $\sim 60\%$ of the dataset simply based on the prompt structure.

Another intriguing result in Table 1 is the comparison between Llama2-13B-Chat and Llama3-8B-Instruct on the Med-HALT dataset. Even though both models have similar accuracies, the ambiguities signal vast differences in the types of errors and underlying behaviour, i.e., significantly high "randomness" for the former. To understand this difference, we map the evaluations to our framework and provide more nuanced results in Figure 3. Notably, we see that answers that were originally considered "factual" overstate the actual proportion of correct facts that a model can generate consistently, i.e., prompt-agnostic factuality. Thus, the true extent of potential harms—both prompt-agnostic errors and randomness together—is greater than what is captured by "hallucination" in existing benchmarks.

### 4.3 DATASET-SPECIFIC TRENDS

We next turn to some dataset-specific trends to highlight insights for future research.

**TruthfulQA.** The TruthfulQA dataset was designed to capture various misconceptions and myths, with carefully crafted adversarial prompts (Lin et al., 2022). Given this construction, it is no surprise that most errors in TruthfulQA are prompt-agnostic, while only a small fraction are attributed to randomness. This makes TruthfulQA an excellent example of a benchmark that can highlight flawed data sources used to train a model. One particularly noteworthy result is the randomness rate of $\sim 10$–$12\%$ across all models, despite their varying accuracy levels. We believe this could be

| | | Detecting Correctness (p-values) | | | | Detecting Consistency (p-values) | | | |
| --- | --- | --- | --- | --- | --- | --- | --- | --- | --- |
| | | Perplexity | Entropy | Surprisal | SelfCheck | Perplexity | Entropy | Surprisal | SelfCheck |
| **Datasets** | TruthfulQA | .89993 | .06291 | .78195 | .06540 | .00003 | .00015 | .02496 | .00031 |
| | Wiki-FACTOR | .03864 | .00003 | .23120 | .00058 | .00003 | .00003 | .00336 | .05768 |
| | Med-HALT | .00003 | .40375 | .00269 | .00288 | .00003 | .00006 | .00833 | .00003 |

Figure 4: Ease of differentiating based on correctness vs consistency, using detection scores.

due to ambiguous questions which require the model to generate 'I have no comment', leading to consistency issues for TruthfulQA.

**Wiki-FACTOR.** The Wiki-FACTOR dataset is constructed using Wikipedia articles, with automatically generated semantically close but adversarial multiple-choice options, thereby increasing the percentage of data points showing randomness (Muhlgay et al., 2024). Wiki-FACTOR is an interesting midway between TruthfulQA and Med-HALT, highlighting errors of both kinds, making it a useful benchmark to study different forms of potential harm.

**Med-HALT.** The Med-HALT dataset combines questions from various medical entrance exams around the world (Pal et al., 2023). We see that it exhibits a significantly higher percentage of randomness compared to other benchmarks. While TruthfulQA demonstrates concerns of unreliable data in training and would require leveraging an external knowledge source to help mitigate errors, Med-HALT represents the other extreme where analyzing the model's uncertainty can be an effective way to detect potential hallucinations.

## 5 HALLUCINATION DETECTION AND MITIGATION

We extend our discussion to existing hallucination detection and mitigation techniques. We provide two key observations: (a) detection techniques primarily differentiate between prompt-agnostic and prompt-sensitive generations rather than identifying factual or hallucinated outputs, and (b) mitigation techniques that rely on knowledge retrieval are themselves influenced by prompt sensitivity, thus introducing additional inconsistencies.

### 5.1 DETECTING CONSISTENCY NOT CORRECTNESS

We start by studying several hallucination detection techniques under our framework. Specifically, we test: (a) **Perplexity**, a simple baseline for hallucination detection (Ren et al., 2022; Chen et al.); (b) **Entropy**, which addresses some of Perplexity's shortcomings (Vashurin et al., 2024); (c) **Surprisal**, based on claims of surprisal using embedding similarity by Duan et al. (2024); and (d) **SelfCheck**, which adapts the intuition behind SelfCheckGPT (Manakul et al., 2023). Each technique produces a final score that can be used to classify the output as a hallucination or not. All detection scores are calculated for the default prompt structure. More details on detection techniques are provided in the appendix (§A.2).

Once we compute the detection scores, we average them separately over all answers labelled as 'correct' and 'incorrect' (for the default prompt structure). This aims to capture the distinction along the axis of correctness. We repeat this across all 16 models in our setup, thus creating a set of 16 average scores for correct answers and the same for incorrect answers. We then test the following hypothesis using the Wilcoxon test: *Assuming the differences in average detection scores for correct and incorrect answers are symmetric around a central value, this central value is zero.* The test aims to determine whether these techniques can easily separate the benchmark into correct and incorrect answers. Finally, we repeat the analysis along the axis of consistency instead of correctness, i.e., averaging detection scores separately for answers labelled as 'prompt-agnostic' and 'prompt-sensitive'.

Note that the detection scores could have more predictive power than measured here, as we're simplifying the distribution into an average. The objective of this test is only to highlight the inherent alignment of detection with consistency, instead of correctness.

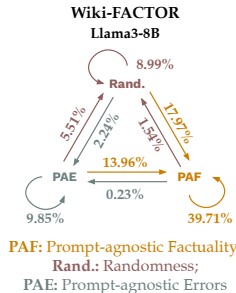

| | **Ambiguity over Retrieved Docs** | | | | | |
| | **Wiki-FACTOR** | | | | **FEVER** | |
| | PAF | PAE | Rand. | PAF | PAE | Rand. |
| GPTNeoX-20B | .27 | .45 | .70 | .87 | .91 | .94 |
| Pythia-12B | .26 | .42 | .75 | .81 | .89 | .93 |
| Bloom-7.1B | .25 | .44 | .73 | .87 | .90 | .95 |
| Llama2-13B-C | .28 | .45 | .69 | .80 | .85 | .91 |
| Llama3-8B-I | .27 | .45 | .73 | .88 | .90 | .93 |
| OPT-30B | .26 | .43 | .74 | .87 | .87 | .95 |

Figure 5: **(Left)** Change in category for questions in the benchmark after the addition of RAG. **(Right)** Randomness under RAG is dominated by the inconsistency of retrieval.

The $p$-values for the hypothesis tests, corresponding to each combination of benchmark and detection technique, are presented in Figure 4. The results reveal a striking pattern: the $p$-values are higher when distinguishing based on correctness, compared to consistency, where the $p$-values are notably low. This shows that the detection techniques are primarily capturing consistency and not correctness, i.e., they are not detecting hallucinations, but instead randomness. This isn't entirely surprising, as many of them are fundamentally designed to detect uncertainty. However, it emphasizes the disconnect between the benchmarks, based on correctness, and the detection techniques, based instead on consistency.

## 5.2 PROMPT SENSITIVITY AND KNOWLEDGE-RETRIEVAL

Most existing hallucination mitigation techniques rely on incorporating external knowledge, typically guided by a retrieval mechanism to find relevant information (Ji et al., 2023). Beyond overall improvements, we find several intriguing shifts in model behaviour during mitigation. Many questions that originally exhibited prompt-agnostic errors instead show randomness, while a smaller portion also shows the opposite trend. Upon deeper investigation, we find that the new component in the pipeline, retrieval, is itself sensitive to prompt variations, introducing an additional layer of inconsistency into the system.

**Retrieval-Augmented Generation (RAG) Setup.** Before jumping into the results, we clarify key details of our RAG setup. We use the in-context retrieval augmentation technique proposed by Ram et al. (2023), using BM25 (Robertson et al., 2009), a sparse word-based retriever. We also leverage the same Wikipedia corpus (Ram et al., 2023) and focus on two datasets: Wiki-FACTOR and FEVER, both originally constructed using facts extracted from Wikipedia. To study the sensitivity of RAG, we introduce paraphrasing variations in the FEVER dataset similar to Wiki-FACTOR, unlike the rest of the paper where we only shuffled demonstrations. Further details about the RAG setup can be found in the appendix (§A.3).

**Results.** We first study the impact of RAG on our evaluations. Figure 5 (Left) captures the movement of all questions in the dataset, as they shift from their original category without RAG to a new category with RAG. The self-loops, thus, indicate questions that remain in the same category. Unsurprisingly, we observe a significant shift towards prompt-agnostic factuality (PAF), i.e., fewer hallucinations. However, a more intriguing result is the redistribution of errors: questions transitioning between prompt-agnostic errors (PAE) and randomness, with an overall flow toward randomness. We argue that this stems from the RAG itself being highly sensitive to prompt changes, thus introducing randomness.

To validate this, we conduct the following test. We extend the idea of 'ambiguity' (Definition 2) to retrieved documents, defining *ambiguity over retrieved documents* as the proportion of questions in the benchmark that can retrieve a different document depending on the prompt structure. Figure 5 (Right) presents these scores across different categories with RAG. The results show that questions exhibiting randomness have significantly higher ambiguity over retrieved documents than others, i.e., the retrieval of different documents for different paraphrasing of the same prompts creates inconsistency. Thus, while RAG can help mitigate factually incorrect generations, it also introduces

its own instability into the pipeline. These results further emphasize the value of evaluating hallucinations within our framework.

# 6 CONCLUSION AND FUTURE WORK

In this paper, we proposed an improved framework for evaluating hallucinations, emphasizing the role of consistency in distinguishing different hallucination harms and informing appropriate detection and mitigation strategies. While our work establishes a strong foundation, several open questions remain. A key challenge is extending to benchmarks that allow freeform generation. Although our fundamental arguments will generalize, the freedom of unconstrained generation introduces new complexities—such as inconsistencies in evaluation setups that rely on LLM judges and redefining self-consistency in the context of language rather than discrete MCQ options. Additionally, future work on different types of prompt variations is also needed. In conclusion, our framework provides a more nuanced approach to hallucination evaluation, allowing the exploration of more effective solutions.

## ACKNOWLEDGMENTS

We thank Florian Carichon and Khaoula Chehbouni for their valuable feedback and comments on earlier drafts of the paper. Funding support for project activities has been partially provided by the Canada CIFAR AI Chair, FRQNT scholarship, and NSERC discovery award. We also express our gratitude to Compute Canada and Mila clusters for their support in providing facilities for our evaluations.

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

## A  ADDITIONAL DETAILS ON EXPERIMENT SETUP

### A.1  DATASETS AND PROMPT VARIATIONS

**TruthfulQA.** We use the MCQ task from TruthfulQA, and adopt the same evaluation setup as used by the original authors (Lin et al., 2022). The evaluation setup contains a 'QA prompt' appended as a prefix, which contains six questions and answers. The original 'QA prompt' can be found in Lin et al. (2022)'s paper. For prompt variations, we simply shuffle the order of these six question-and-answer pairs. We measure all metrics across 50 different prompt variations, i.e., 50 unique shufflings of these pairs.

**Wiki-FACTOR.** Instead of using the complete prefix from the Wiki-FACTOR dataset, we instead use only the shorter 'context' (Muhlgay et al., 2024). Since the Wiki-FACTOR dataset has no prompt

template, we have to rely on paraphrasing to introduce prompt variations. We use the fine-tuned T5-based paraphraser as mentioned in the main text (Vorobev & Kuznetsov, 2023a;b). We measure all metrics across 10 different prompt variations, i.e., 10 different paraphrases of the prompt.

**Med-HALT.** We use the Reasoning Hallucination Test (RHT) of the Med-HALT dataset, and the original instruction prompt used by the authors (Pal et al., 2023). However, we do not form the problem as a reasoning test. Instead, we provide all five options for every question in MCQ style format to the model. Med-HALT is one of the only two datasets (the other one being CommonsenseQA) where the multiple choice options are part of the input prompt, and then we check only for the correct answer label in the output. For prompt variations, we shuffle the ordering of options for MCQ. We measure all metrics across 20 different prompt variations, i.e., 20 different shufflings of the MCQ options.

**CommonsenseQA.** We use the development set of the dataset and perform a 16-shot evaluation of the CommonsenseQA benchmark. The formatting of each question is the same as the Med-HALT dataset, i.e., the MCQ options are given as part of the input prompt. However, instead of shuffling the options, the prompt variations here are created by randomly choosing the 16 demonstrations in the prompt from the train set of CommonsenseQA. We measure all metrics across 50 different prompt variations, i.e., 50 different random choices of the 16-shot demonstrations.

**FEVER.** We use the shared task development set of the dataset and perform a 16-shot evaluation of the FEVER benchmark. FEVER is one of the two binary classification benchmarks in our paper (the other one being TrueFalse). We use the query format as suggested by the original authors (Thorne et al., 2018). Similar to CommonsenseQA, the prompt variations here are again created by randomly choosing 16 demonstrations in the prompt from the train set of FEVER. We measure all metrics across 50 different prompt variations, i.e., 50 different random choices of the 16-shot demonstrations.

**TrueFalse.** We use all topics combined from the TrueFalse dataset and perform a 16-shot evaluation of the benchmark. We use the same query format as FEVER (Thorne et al., 2018). Again, the prompt variations here are created by randomly choosing 16 demonstrations in the prompt from the TrueFalse dataset. There is no separate train set to sample from and hence the demonstrations are sampled from the evaluation dataset itself. Thus, the sampled demonstration in certain cases might even contain the final question. We measure all metrics across 50 different prompt variations, i.e., 50 different random choices of the 16-shot demonstrations.

## A.2 Detection Techniques

We provide details on the scores calculated for all four detection techniques in our paper.

**Perplexity.** We simply use the length-normalized perplexity score of the best option.

**Entropy.** We treat the length-normalized perplexity scores of all options as scores of a classification problem, and measure the entropy of the prediction. In other words, we first normalize the perplexity scores across all options to turn them into probabilities, and then measure the entropy of the probabilities across all options.

**Surprisal.** We measure the cosine similarity between the representation of the question and the representation of the chosen option appended to the question, as suggested by Duan et al. (2024). Here, the representation of a sentence is the output of the final transformer layer of the model, i.e., the final hidden state, for the last token.

**SelfCheck.** Adopting from Manakul et al. (2023), we simply append the chosen option to the question, followed by a follow-up question 'Is the above statement correct?', and check for the probability of the next token being 'Yes'.

## A.3 Mitigation Setup Details

**RAG Setup.** We use the open-source code provided by Ram et al. (2023)[2].

---

[2]`https://github.com/AI21Labs/in-context-ralm`

| | TruthfulQA | | Wiki-FACTOR | | Med-HALT | |
|---|---|---|---|---|---|---|
| | Accuracy (%) | Ambiguity (%) | Accuracy (%) | Ambiguity (%) | Accuracy (%) | Ambiguity (%) |
| GPTJ-6B | $22.86_{\pm 0.71}$ | 13.83 | $41.98_{\pm 0.90}$ | 39.65 | $28.99_{\pm 0.77}$ | 50.17 |
| GPTNeoX-20B | $20.09_{\pm 1.26}$ | 22.89 | $45.74_{\pm 1.38}$ | 41.95 | $28.98_{\pm 0.43}$ | 52.26 |
| Pythia-2.8B | $23.37_{\pm 1.20}$ | 16.65 | $37.93_{\pm 0.84}$ | 40.21 | $28.21_{\pm 0.70}$ | 49.78 |
| Pythia-6.9B | $21.99_{\pm 1.19}$ | 13.95 | $40.87_{\pm 0.89}$ | 39.08 | $28.39_{\pm 0.42}$ | 37.63 |
| Pythia-12B | $20.31_{\pm 0.89}$ | 19.58 | $42.90_{\pm 0.96}$ | 38.61 | $28.18_{\pm 0.39}$ | 50.07 |
| Bloom-3B | $25.56_{\pm 1.18}$ | 16.16 | $30.27_{\pm 0.83}$ | 38.58 | $27.95_{\pm 1.42}$ | 70.07 |
| Bloom-7.1B | $23.18_{\pm 1.32}$ | 18.36 | $35.14_{\pm 0.78}$ | 37.27 | $28.51_{\pm 0.62}$ | 56.40 |
| Llama2-7B | $25.65_{\pm 0.73}$ | 16.16 | $47.87_{\pm 1.32}$ | 38.81 | $34.00_{\pm 0.65}$ | 61.79 |
| Llama2-7B-C | $31.11_{\pm 0.79}$ | 19.34 | $45.25_{\pm 1.05}$ | 47.70 | $33.56_{\pm 1.15}$ | 70.14 |
| Llama2-13B | $27.76_{\pm 0.66}$ | 17.26 | $52.41_{\pm 1.52}$ | 41.08 | $37.57_{\pm 0.21}$ | 58.00 |
| Llama2-13B-C | $32.97_{\pm 1.03}$ | 21.79 | $50.32_{\pm 1.06}$ | 47.09 | $34.84_{\pm 0.42}$ | 60.54 |
| Llama3-8B | $28.85_{\pm 1.16}$ | 18.48 | $52.69_{\pm 1.54}$ | 40.25 | $40.06_{\pm 0.61}$ | 48.28 |
| Llama3-8B-I | $39.34_{\pm 0.75}$ | 17.14 | $48.39_{\pm 1.19}$ | 42.32 | $34.55_{\pm 0.23}$ | 31.03 |
| OPT-6.7B | $22.26_{\pm 0.92}$ | 15.42 | $39.58_{\pm 1.04}$ | 38.81 | $28.20_{\pm 0.78}$ | 51.22 |
| OPT-13B | $21.81_{\pm 1.11}$ | 19.71 | $41.34_{\pm 1.04}$ | 42.59 | $28.30_{\pm 0.53}$ | 43.86 |
| OPT-30B | $22.55_{\pm 0.90}$ | 23.38 | $43.58_{\pm 0.91}$ | 41.35 | $28.32_{\pm 0.42}$ | 50.00 |

Table 2: Extended results across all models of Table 1.

**FEVER Dataset.** The FEVER dataset contains several demonstrations that are shuffled to create prompt variations, as mentioned above. However, for the mitigation portion of the paper, we wanted to highlight the impact of prompt paraphrasing on the retrieval component, and hence, we changed the way prompt variations were created for FEVER. While we still provide the demonstrations, we do not shuffle them and instead paraphrase the question the same way as we did for the Wiki-FACTOR dataset, using the same T5-based paraphrase. Similar to Wiki-FACTOR, we create 10 variations, i.e., 10 different paraphrases of each prompt.

## B  EXTENDED RESULTS FOR TABLES IN THE MAIN PAPER

Several results in the main text were reported only for a few models, and we extend the rest of the results here. Extended results for Table 1 are present in Table 2 and extended results for Figure 5 (Right) are present in Table 3. The trends in these models are still similar to the trends in the main paper.

## C  RESULTS ON COMMONSENSEQA, FEVER, AND TRUEFALSE

Additional results on CommonsenseQA, FEVER, and TrueFalse datasets are in Table 4 and Figure 6. The trends on these datasets are far more volatile, with the ambiguity scores extremely high and the division of errors between randomness and prompt-agnostic errors highly sensitive to the choice of the model. Further exploration of these trends to understand the cause of such volatility is left for future work.

| | Ambiguity over Retrieved Docs | | | | | |
| | Wiki-FACTOR | | | FEVER | | |
| | PAF | PAE | Rand. | PAF | PAE | Rand. |
|---|---|---|---|---|---|---|
| GPT-J-6B | .26 | .43 | .74 | .86 | .87 | .95 |
| GPTNeoX-20B | .27 | .45 | .70 | .87 | .91 | .94 |
| Pythia-2.8B | .25 | .45 | .75 | .87 | .88 | .95 |
| Pythia-6.9B | .26 | .45 | .74 | .87 | .89 | .95 |
| Pythia-12B | .26 | .42 | .75 | .81 | .89 | .93 |
| Bloom-3B | .24 | .45 | .72 | .92 | .91 | .88 |
| Bloom-7.1B | .25 | .44 | .73 | .87 | .90 | .95 |
| Llama-2-7B | .28 | .44 | .74 | .90 | .89 | .92 |
| Llama-2-7B-Chat | .27 | .44 | .70 | .90 | .89 | .92 |
| Llama-2-13B | .29 | .44 | .73 | .90 | .90 | .91 |
| Llama-2-13B-C | .28 | .45 | .69 | .80 | .85 | .91 |
| Llama-3-8B | .29 | .46 | .70 | .89 | .91 | .94 |
| Llama-3-8B-I | .27 | .45 | .73 | .88 | .90 | .93 |
| OPT-6.7B | .26 | .44 | .74 | .89 | .89 | .93 |
| OPT-13B | .27 | .42 | .72 | .92 | .90 | .89 |
| OPT-30B | .26 | .43 | .74 | .87 | .87 | .95 |

Table 3: Extended results across all models of Figure 5 (Right).

| | CommonsenseQA | | FEVER | | TrueFalse | |
| | Accuracy (%) | Ambiguity (%) | Accuracy (%) | Ambiguity (%) | Accuracy (%) | Ambiguity (%) |
|---|---|---|---|---|---|---|
| GPT-J-6B | $36.55_{\pm0.70}$ | 81.16 | $57.47_{\pm3.62}$ | 71.31 | $51.05_{\pm3.11}$ | 100.00 |
| Pythia-2.8B | $26.19_{\pm0.84}$ | 75.59 | $52.48_{\pm3.35}$ | 58.39 | $51.34_{\pm3.13}$ | 100.00 |
| Pythia-6.9B | $25.27_{\pm0.63}$ | 79.93 | $57.73_{\pm3.69}$ | 82.57 | $49.28_{\pm2.82}$ | 100.00 |
| Pythia-12B | $31.88_{\pm0.94}$ | 81.82 | $51.85_{\pm2.01}$ | 20.60 | $53.90_{\pm5.38}$ | 99.89 |
| Bloom-3B | $28.41_{\pm1.22}$ | 87.14 | $57.34_{\pm4.00}$ | 89.54 | $48.95_{\pm2.32}$ | 100.00 |
| Bloom-7.1B | $30.32_{\pm0.90}$ | 82.31 | $50.03_{\pm0.06}$ | 0.61 | $50.22_{\pm2.85}$ | 100.00 |
| Llama2-7B | $68.18_{\pm0.73}$ | 48.40 | $53.37_{\pm4.22}$ | 54.06 | $77.40_{\pm9.42}$ | 65.75 |
| Llama2-7B-C | $69.28_{\pm0.67}$ | 48.48 | $62.73_{\pm6.17}$ | 52.93 | $79.87_{\pm6.72}$ | 40.87 |
| Llama2-13B | $73.78_{\pm0.49}$ | 35.30 | $51.34_{\pm2.54}$ | 11.51 | $82.45_{\pm9.03}$ | 45.43 |
| Llama2-13B-C | $73.95_{\pm0.63}$ | 38.49 | $64.44_{\pm8.09}$ | 44.66 | $87.26_{\pm2.57}$ | 27.28 |
| Llama3-8B | $74.03_{\pm0.53}$ | 34.89 | $57.23_{\pm11.91}$ | 44.11 | $92.01_{\pm2.64}$ | 18.72 |
| Llama3-8B-I | $78.26_{\pm0.49}$ | 31.70 | $81.53_{\pm2.29}$ | 34.04 | $92.79_{\pm0.84}$ | 12.79 |
| OPT-6.7B | $27.41_{\pm0.86}$ | 95.33 | $55.47_{\pm3.50}$ | 99.03 | $51.85_{\pm3.66}$ | 100.00 |
| OPT-13B | $30.97_{\pm0.88}$ | 88.70 | $53.09_{\pm1.85}$ | 98.23 | $51.27_{\pm4.23}$ | 98.29 |

Table 4: Additional results for ambiguity scores.

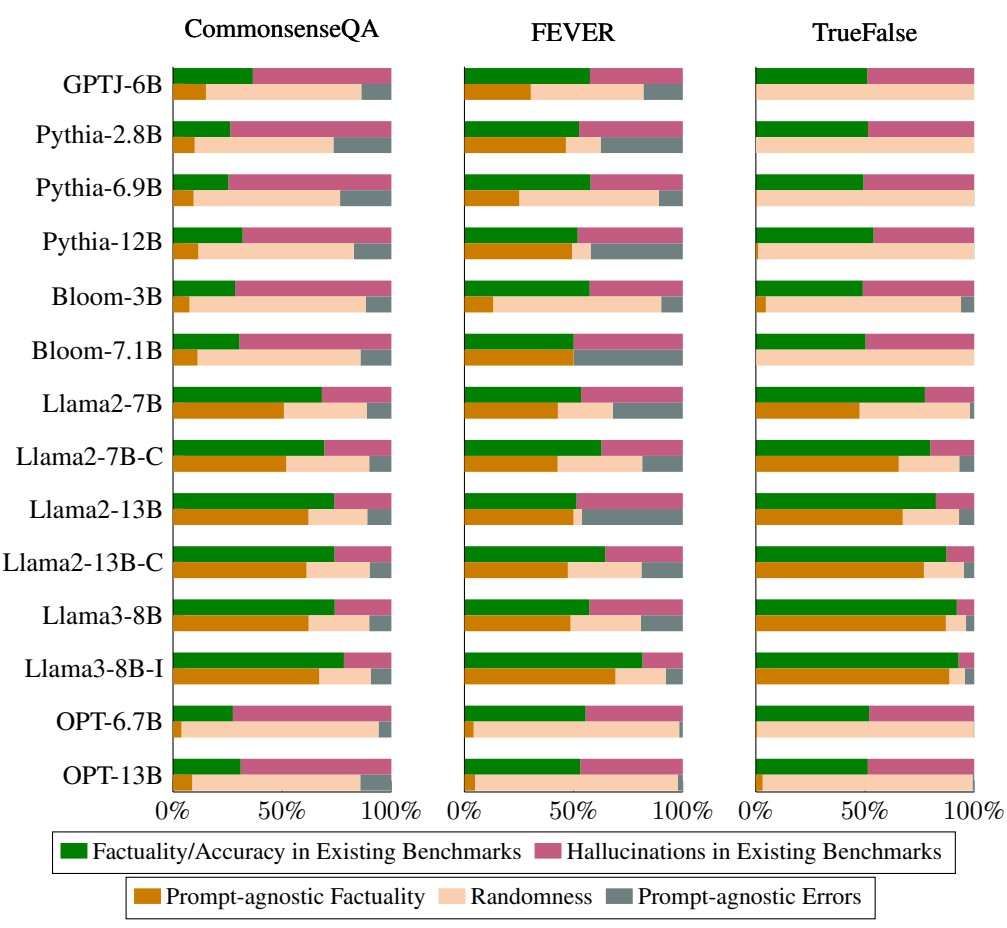

Figure 6: Additional LLM hallucination benchmark results under our new framework.

