# OpenReview forum: "Rethinking Hallucinations: Correctness, Consistency, and Prompt Multiplicity"
_ICLR.cc/2025/Workshop/BuildingTrust — BuildingTrust_

### Official Review · Reviewer_3FUh · 2025-02-22
**Well-written, contribution is minor**

**Rating:** 5
**Confidence:** 3

**Review:**

# Summary
This work examines the influence of prompt formulation on LLM hallucinations. They predominantly focus on shuffling to vary prompts.
The paper introduces prompt multiplicity, which measures the output agreement (on a given question) across different prompt formulations. Using this approach, the authors break down errors made by LLMs into prompt-sensitive and prompt-agnostic.

# Review:
Overall, I'm unsure about this paper. The paper is generally well-written, but the contribution is minor.

The observation that prompt design can influence which questions a model answers correctly—while maintaining the same overall accuracy—is interesting. The main difference with existing literature (on the influence of prompt design on accuracy) is the level of analysis. This work focuses on the question-level, whereas existing work focuses more on the (aggregate) dataset-level.

However, I expected the authors to suggest a way to leverage this insight— e.g., show how multiplicity could be used to improve performance or as a safety mechanism. Instead, they suggest using multiplicity to break down model performance. Concretely, they suggest breaking down correct answers into “prompt-agnostic factuality” and randomness. This measures whether the model is robust to prompt variations. While interesting, this does seem close to existing work (see implementation comments).
Next, they recommend breaking down incorrect answers into prompt-agnostic errors and randomness (prompt-sensitive errors). This distinguishes the cases in which the model consistently outputs an incorrect answer, or outputs different (but still incorrect) answers.
As mentioned by the authors, uncertainty captures this axes as it can be used to capture 'randomness' vs non-random answers.

As acknowledged by the authors, this means that the main contribution of this framework is the distinction between prompt-agnostic factuality and prompt-agnostic errors. However, this breakdown is similar to the classic notions of ‘correctness’ and ‘robustness’.

My key takeaway from the paper is that prompts influence which questions a model answers correctly. However, there is not a *predictable* way in which the prompt influences the correctness (e.g., a proposed style or aspect). Therefore, it becomes more of a claim about robustness. While this is an interesting observation, I would have liked to see a more practical application of this insight.

Implementation comments:
- The authors measure multiplicity using shuffling.  Lu et al. (https://arxiv.org/abs/2104.08786, as cited by the authors) specifically examine how factuality is affected by shuffling, which diminishes the novelty of this contribution.
- Table 2: In the uncertainty community, the use of probabilities (or perplexity) has been criticized. The relationship between perplexity and sensitivity is clear, and therefore this result does not add as much. Instead, it would be more appropriate to report predictive entropy.

# Minor details and suggestions
- Definition 1/2 -- should it be $y_i \in \mathbb{Z}+^n$?
- I'd recommend placing definition 5 before definition 3/4 — it makes it easier for the reader.
- A more in-depth discussion of the uncertainty literature would be appropriate -- some of the key ideas here (e.g., disagreement across different prompt variations) relate to ideas in the classic uncertainty literature. For example, the same disagreement is leveraged in ensembles (https://arxiv.org/abs/1612.014740).
- Please state what value for $\tau$ is used in the main text. It'd also be nice to see how the results change for different values of $\tau$.

---

### Official Review · Reviewer_rVrC · 2025-03-02

**Rating:** 8
**Confidence:** 4

**Review:**

This paper makes a significant contribution to our understanding of hallucinations in LLMs by introducing the concept of "prompt multiplicity" - how variations in prompts can cause models to switch between correct and incorrect answers despite maintaining similar overall accuracy scores. The authors conduct extensive experiments across multiple benchmarks and model families to demonstrate that existing hallucination evaluation methods fail to capture this important dimension of model reliability.

Strengths:
* The research formalizes prompt multiplicity as a metric for evaluating hallucination stability, building on existing multiplicity literature and adapting it specifically for LLM evaluation.
* It features a comprehensive experimental design covering 6 different benchmarks and 16 models across various families.
* The authors propose a refined taxonomy that separates hallucinations into "prompt-agnostic errors" (consistent factual mistakes) and "randomness" (inconsistent answers due to prompt sensitivity), which helps better understand the different types of potential harms.
* The research demonstrates that existing uncertainty-based hallucination detection techniques primarily identify randomness rather than factual errors, highlighting the limitations of current approaches.


Weaknesses:
* The study relies exclusively on MCQ-style benchmarks. Extending this framework to different benchmark types, particularly those involving free-form generation, would provide more comprehensive insights into prompt multiplicity across diverse task settings.
* Most variations are created simply by shuffling demonstrations. A more systematic investigation into how demonstration quality and location affect hallucination rates would likely reveal stronger factors influencing prompt multiplicity than mere ordering.
* The MCQ format significantly reduces the potential for "randomness" by limiting the output to essentially a single token from a small set of choices. This artificial constraint likely underestimates the extent of randomness-based hallucinations that would occur in more open-ended, realistic use cases where models generate unrestricted text.

---

### Decision · Program_Chairs · 2025-03-04

Accept